# Distinct polymorphisms in a single herpesvirus gene are capable of enhancing virulence and mediating vaccinal resistance

**Andelé M. Conradie**[1], **Luca D. Bertzbach**[1], **Jakob Trimpert**[1], **Joseph N. Patria**[2], **Shiro Murata**[3], **Mark S. Parcells**[4], **Benedikt B. Kaufer**[1]*

1 Institut für Virologie, Freie Universität Berlin, Berlin, Germany, 2 Department of Biological Sciences, University of Delaware, Newark, United States of America, 3 Department of Disease Control, Faculty of Veterinary Medicine, Hokkaido University, Sapporo, Japan, 4 Department of Animal and Food Sciences, University of Delaware, Newark, United States of America

* b.kaufer@fu-berlin.de

**Data Availability Statement:** All relevant data are within the manuscript and its Supporting Information files.

## Abstract

Modified-live herpesvirus vaccines are widely used in humans and animals, but field strains can emerge that have a higher virulence and break vaccinal protection. Since the introduction of the first vaccine in the 1970s, Marek's disease virus overcame the vaccine barrier by the acquisition of numerous genomic mutations. However, the evolutionary adaptations in the herpesvirus genome responsible for the vaccine breaks have remained elusive. Here, we demonstrate that point mutations in the multifunctional *meq* gene acquired during evolution can significantly alter virulence. Defined mutations found in highly virulent strains also allowed the virus to overcome innate cellular responses and vaccinal protection. Concomitantly, the adaptations in *meq* enhanced virus shedding into the environment, likely providing a selective advantage for the virus. Our study provides the first experimental evidence that few point mutations in a single herpesviral gene result in drastically increased virulence, enhanced shedding, and escape from vaccinal protection.

## Author summary

Viruses can acquire mutations during evolution that alter their virulence. An example of a virus that has shown repeated shifts to higher virulence in response to more efficacious vaccines is the oncogenic Marek's disease virus (MDV) that infects chickens. Until now, it remained unknown which mutations in the large virus genome are responsible for this increase in virulence. We could demonstrate that very few amino acid changes in the *meq* oncogene of MDV can significantly alter the virulence of the virus. In addition, these changes also allow the virus to overcome vaccinal protection and enhance the shedding into the environment. Taken together, our data provide fundamental insights into evolutionary changes that allow this deadly veterinary pathogen to evolve towards greater virulence.

**Funding:** This research was funded by the Volkswagen Foundation Lichtenberg grant A112662 awarded to B.B.K. The funders had no role in study design, data collection and analysis, decision to publish, or preparation of the manuscript. Authors, A.M.C and L.D.B received a salary from the funding source. The authors, J.T., J.N.P, S.M., M.S.P, received no specific funding for this work.

**Competing interests:** The authors have declared that no competing interests exist.

## Introduction

Vaccines have revolutionized modern medicine and industrial animal farming by dramatically lowering disease incidence and mortality [1,2]. While vaccines are ideal interventions for eradication, some viruses can evolve to overcome vaccinal protection [3]. Therefore, it is crucial to understand the evolutionary changes that facilitate vaccine resistance in order to develop more effective vaccines. [4]. A well-documented example of virus evolution towards a greater virulence is the highly oncogenic Marek's disease virus (MDV) [5,6]. MDV is an alphaherpesvirus that infects chickens and is controlled by the wide application of modified live virus vaccines. In the absence of vaccination, infected chickens typically develop an acute rash, and edematous neuronal and brain damage, severe lymphomas, paralysis, and death at a very young age [7,8]. The tumors induced by MDV are considered to be one of the most frequent cancers in the animal kingdom [9].

MDV has undergone three major shifts in virulence over the past decades (Fig 1A). This evolution resulted in ever more virulent field strains that cause increased severe clinical symptoms and vaccine evasion [8,10,11]. MDV strains are currently classified into four pathotypes based on their pathogenicity in vaccinated and unvaccinated chickens [8,12,13]. First-generation MDV vaccines, such as the related herpesvirus of turkey (HVT), were introduced in the 1970s to prevent chickens from emerging virulent MDV (vMDV) strains [14]. Soon after the introduction of the HVT vaccine, very virulent (vvMDV) strains emerged that were more pathogenic, immunosuppressive, and were able to overcome this vaccinal protection [15]. Protection against vvMDV was achieved using a second-generation bivalent vaccine, composed of a combination of a non-oncogenic, related herpesvirus of chickens (MDV-2, strain SB1) with HVT that protected chickens from clinical disease [14]. Subsequently, very virulent plus (vv+MDV) strains emerged that are controlled by the third-generation vaccine (CVI988/Rispens); however, it remains unknown if more virulent strains will arise in the future (Fig 1A) [14,16]. This stepwise evolution of MDV directly correlates with the introduction of MD vaccines [17], suggesting that the 'leaky' MDV vaccines that protect from disease but are unable to provide sterilizing immunity may have directly contributed to the increase in virulence [18].

A large number of MDV field strains from all pathotypes have been sequenced over the years to identify mutations that could be responsible for changes in virulence [19,20]. A few defined point mutations in the coding sequence of the major MDV oncogene *meq* have been identified that coincide with increased virulence (Fig 1A) [10,20]; however, their contribution in the evolution of MDV towards a greater virulence has never been proven.

Meq is a 339 amino acid basic leucine zipper protein (bZIP) that is expressed in lytically and latently infected cells, and is encoded in the internal and terminal repeat regions of the MDV genome [21]. Meq regulates viral and cellular genes by forming heterodimers with other bZIP proteins such as c-Jun to promote transcription [22]. In addition, Meq can form homodimers that repress the expression of numerous genes [22–25]. The C-terminus of *meq* encodes a transactivation domain characterized by proline-rich repeats (PRR) [26]. Low virulent vMDV strains (e.g. JM/102W) contain five PRR in their C-terminus, whereas vvMDV (e.g. RB-1B) and vv+MDV strains (e.g. N-strain) possess only three PRR (Fig 1A) [27].

In this study, we set out to determine if these point mutations acquired in *meq* through the years contribute to the increase in MDV virulence, vaccine resistance and virus transmission. The *meq* isoforms of different pathotypes (vMDV, vvMDV, vv+MDV and the CVI988/Rispens vaccine strain) were individually inserted into the very virulent RB-1B strain, thereby replacing its original *meq* gene. Virus replication was not significantly affected *in vitro* and *in vivo*. However, insertion of less virulent *meq* isoforms (vacMeq and vMeq) either abrogated or severely impaired MDV pathogenesis while higher virulent *meq* isoforms (vvMeq and vv+Meq) readily

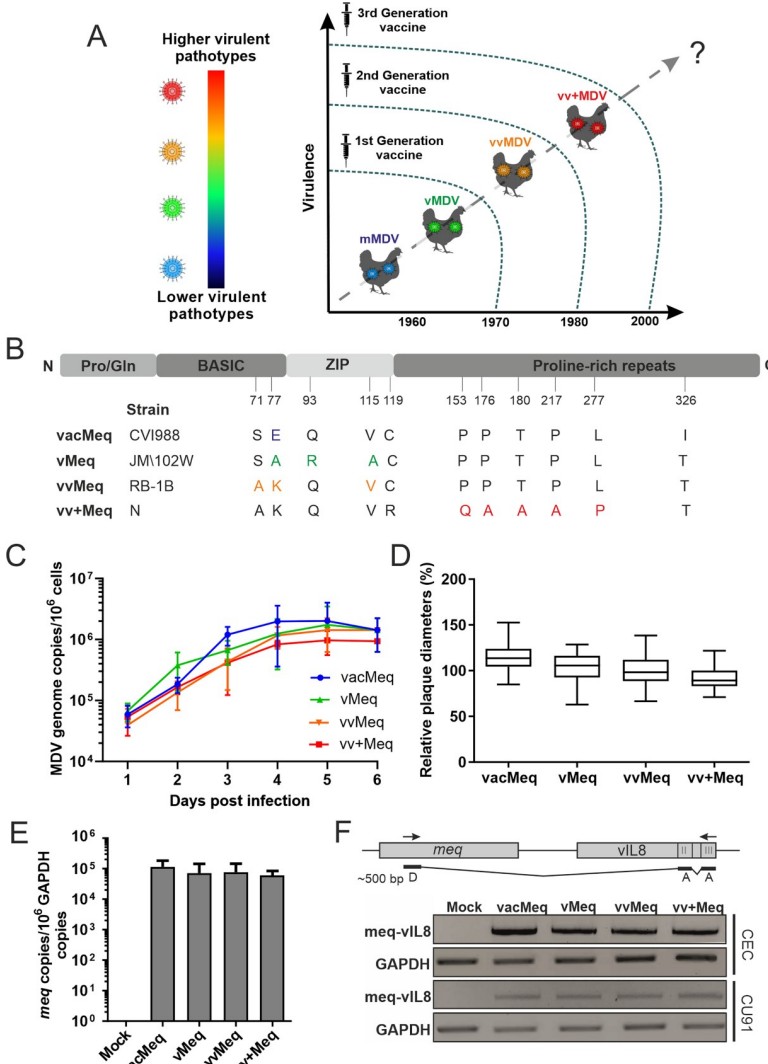

**Fig 1. Characterization of the recombinant viruses *in vitro*.** (A) A schematic illustration of the evolution of MDV towards increased virulence in the context of the indicated vaccine generations. (B) The representation of the Meq protein with its domains. The N-terminal region comprises of a proline/glutamine (Pro/Gln) rich domain followed by the basic region and the leucine zipper (ZIP). (C) Virus replication was assessed by multi-step growth kinetics. Mean viral genome copies per one million cells are shown for the indicated viruses and time points (p>0.05, Kruskal–Wallis test, n = 3). (D) Plaque size assays of indicated recombinant viruses. The mean plaque diameters of three independent experiments are shown as box plots with minimums and maximums (p>0.05, one-way ANOVA, n = 150). (E) The *meq* expression levels in infected CEC were assessed by RT-qPCR. *Meq* expression is shown relative to one million copies of the cellular glyceraldehyde-3-phosphate dehydrogenase (GAPDH) and were not statistically different (Kruskal-Wallis test). (F) RT-PCR analysis of the *meq*/vIL8 splice variant using primers specific for the donor site "D" in *meq* and the acceptor sites "A" in vIL8. GAPDH was used as a control.

caused disease and tumors. Even in vaccinated chickens, viruses harboring the higher virulent *meqs* caused disease and efficiently shed into the environment. Strikingly, only viruses harboring the vv+Meq were able to overcome vaccinal protection and cause tumors in vaccinated animals. Furthermore, we show that the point mutations in *meq* isoforms of higher virulent MDV strains help the virus to overcome innate cellular responses, potentially contributing to vaccine failure. Overall, our data show that the evolutionary adaptations in *meq* substantially

contribute to the increased virulence, vaccine resistance, and enhanced transmission–therefore playing a central role in the evolution of this highly oncogenic alphaherpesvirus.

## Results

### Generation of recombinant viruses

To determine if the point mutations in the *meq* isoforms contribute to MDV evolution towards a greater virulence, we replaced the *meq* gene in the very virulent RB-1B MDV strain with the *meqs* from different pathotypes as described previously [28]. Briefly, the *meq* gene from the CVI988/Rispens vaccine strain, JM/102W (vMDV), RB-1B (vvMDV) or N-strain (vv +MDV) were inserted into a virus lacking the *meq* gene (Δmeq) [28] by two-step Red-mediated mutagenesis [29,30]. The insertion of *meq isoforms* were confirmed by next-generation sequencing (S2A Fig). The recovered recombinant viruses were termed vacMeq, vMeq, vvMeq and vv+Meq. Sequencing of the recombinant viruses, passage level 4, confirmed the presence of the respective *meq* isoforms in the $TR_L$ and $IR_L$ without any secondary mutations in the genome (S2B Fig) or in the *meq* genes (S2 Table).

### Characterization of recombinant viruses *in vitro*

To determine if the *meq* isoforms of different pathotypes affect virus replication, we performed plaque size assays and demonstrated that all viruses efficiently replicated *in vitro*, while minor changes were observed that were not statistically significant. The *meq* genes from less virulent strains slightly enhanced replication *in vitro* (Fig 1C), a phenotype also observed with the corresponding parental strains [31]. We confirmed this phenotype by plaque size assays (Fig 1D), underlining that the insertion of *meq* isoforms only mildly affects MDV replication. We verified that all *meq* isoforms are expressed at comparable levels by performing RT-qPCR on samples from infected chicken embryo cells (CEC) (Fig 1E). Furthermore, we analyzed whether the splice variant of *meq* to exons II and exons III of vIL8 (*meq*/vIL8) is affected through the differences in *meq*. Our data revealed the *meq*/vIL8 splicing is not affected in CEC and CU91 T cells (Fig 1F), which is consistent with the absence of changes in the splice sites.

### Role of the *meq* isoforms in MDV pathogenesis

To investigate if the evolutionary acquired point mutations in the *meq* gene contribute to MDV-induced pathogenesis and tumor formation, one-day old unvaccinated chickens were infected subcutaneously with 4,000 pfu of the respective recombinant viruses. To determine the effect of the *meq* isoforms on MDV replication, we quantified viral genome copies in the blood of infected animals by qPCR. All viruses efficiently replicated in infected animals (Fig 2A), indicating that the changes in the *meq* isoforms only have a minor contribution to lytic replication *in vivo*. We monitored the animals for clinical disease symptoms and tumors during the experiment. Replacement with the MDV vaccine *meq* isoform completely abrogated virus-induced pathogenesis and tumor formation (Fig 2B and 2C). Viruses harboring the vMDV *meq* isoform only induced clinical disease in 20% of the animals, while only 10% developed gross tumors (Fig 2B and 2C). vvMeq and vv+Meq efficiently induced disease and tumors, while the native vvMeq resulted in the highest virulence (Fig 2B and 2C).

To assess the effect of the *meq* isoforms on tumor dissemination, the number of visceral organs with macroscopic tumors were quantified during necropsy throughout the course of the experiment and at the day of final necropsy (86 dpi). Replacement with the vMDV *meq* severely impaired tumor dissemination (Fig 2D), as only a single organ (spleen) was affected in each tumor-bearing animal. vvMeq and vv+Meq induced efficient tumor dissemination in

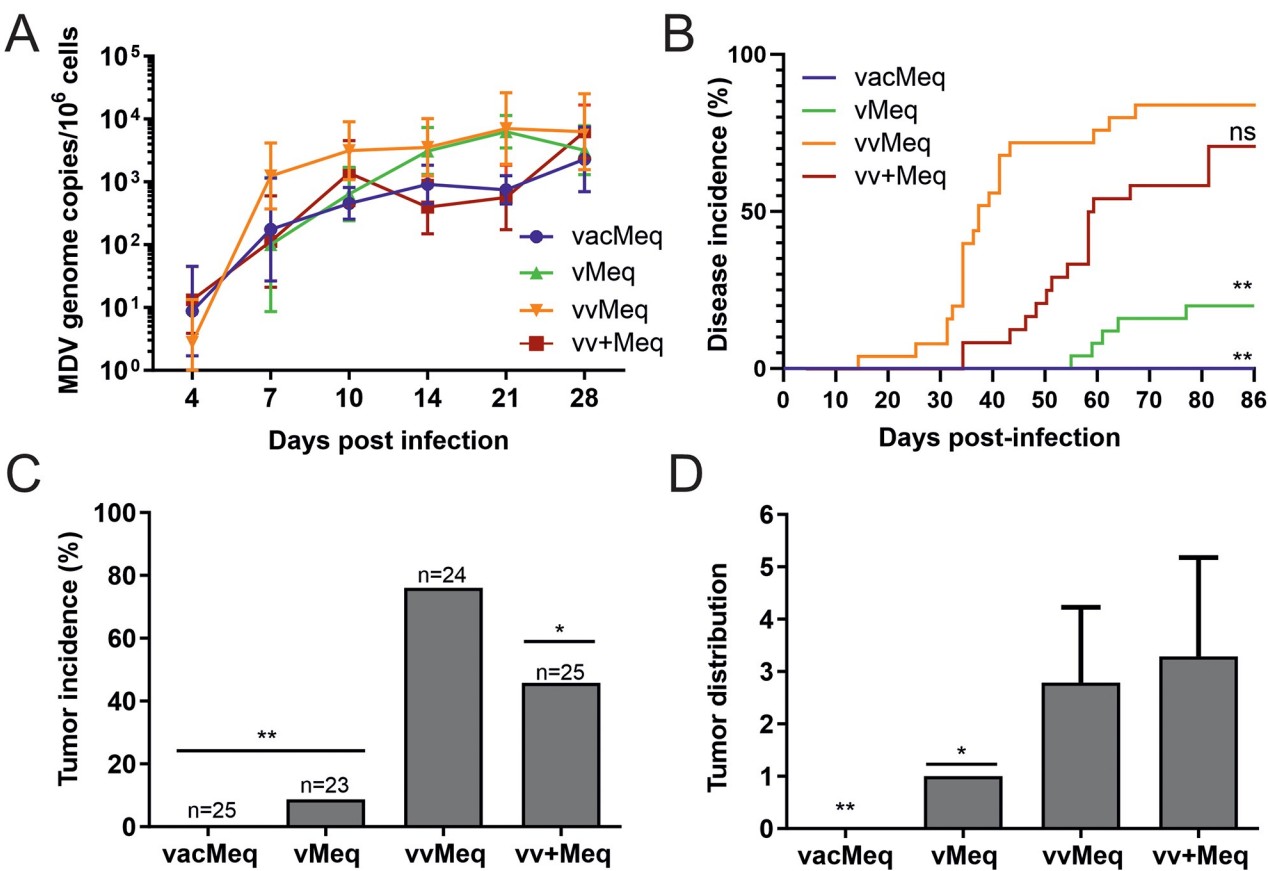

**Fig 2. Influence of *meq* isoforms from various pathotypes on MDV pathogenesis.** (A) MDV genome copies were detected in the blood samples of chickens infected with indicated viruses by qPCR. Mean MDV genome copies per one million cells are shown for the indicated time points ($p > 0.05$, Kruskal-Wallis test). (B) Disease incidence in chickens infected with indicated recombinant viruses and significant differences in comparison to vvMeq (** $p < 0.0125$, Log-rank (Mantel-Cox) test). (C) Tumor incidence as percentage of animals that developed tumors during the experiment. Asterisks indicate significant differences compared to vvMeq (* $p < 0.05$ and ** $p < 0.0125$; Fisher's exact test). (D) Tumor distribution is shown as the number of tumorous organs in tumor-bearing animals with standard deviations (* $p < 0.05$ and ** $p < 0.0125$; Fisher's exact test).

contrast to the lower virulent *meq* isoforms (Fig 2D). The data of this in vivo experiment was validated in an independent animal experiment using a different chicken line. In this second animal experiment, we observed a comparable MD incidence and tumor incidence (S1 Fig). To ensure that the viruses did not develop compensatory mutations in the animals, we performed next-generation sequencing on viruses derived from organs and tumors (n = 12). Most viruses did not acquire any mutations in the animals, while three viruses had a single mutation that was either silent or in a non-coding region (S2C Fig). In addition, we confirm that the *meq* was not altered in the host (S2C Fig). These experiments revealed that the mutations in the *meq* isoforms affect virus-induced pathogenesis, tumor formation, and dissemination.

## Natural spread and pathogenesis of recombinant viruses in contact animals

To confirm that these effects are also observed upon the natural spread of the virus via the respiratory tract, we co-housed naïve chickens with the subcutaneously infected animals. All *meq* isoform viruses were readily transmitted to the contact chickens as viral copies were detected in the blood (Fig 3A), but only viruses harboring the vv and vv+ *meq* isoforms caused disease (Fig 3B). Insertion of *meq* isoforms from the CVI988/Rispens vaccine and vMDV pathotypes completely abrogated tumor formation (Fig 3C). Viruses harboring the vvMDV

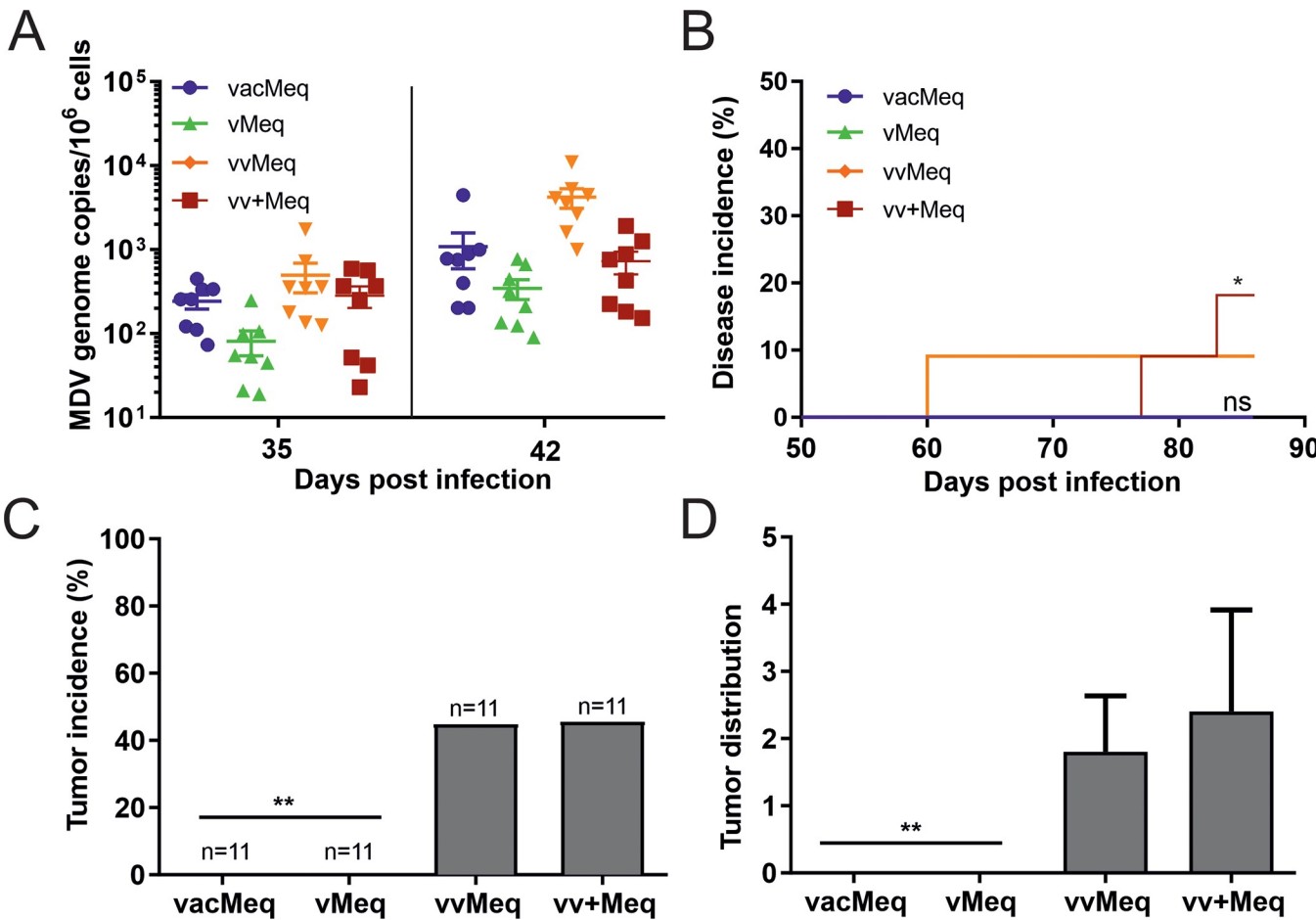

**Fig 3. Pathogenesis and tumor incidence in naïve contact animals.** (A) qPCR analysis of blood samples from naive chickens where MDV genome copies were determined (p>0.05, Kruskal-Wallis test). (B) Disease incidence in naïve chickens infected via the natural route and tumor incidence (C) and tumor distribution (D) are shown for co-housed contact animals. Asterisks (** p<0.0125; Fisher's exact test) indicate the significant differences in (C).

and vv+MDV *meq* isoforms both efficiently induced tumors in the contact animals. As observed in the subcutaneously infected animals, tumor dissemination of the vv+Meq was slightly enhanced, although not statistically different, compared to the very efficient vvMeq (Fig 3D).

Our data demonstrate that the few point mutations in the *meq* gene directly contribute to MDV virulence in experimentally and naturally infected animals.

## Pathogenesis of *meq* isoforms in vaccinated animals

Next, we determined if the different *meq* isoforms contribute to vaccine resistance and affect virus shedding in vaccinated animals. One-day old chickens were vaccinated subcutaneously with 4,000 pfu of the commonly used HVT vaccine. At seven days post-vaccination, we infected all vaccinated chickens with 5,000 pfu of the respective recombinant viruses to determine if *meq* contributes to vaccine breaks. Replication of the recombinant viruses (Fig 4A) and HVT vaccine (Fig 4B) was not statistically different between the groups. Vaccination completely protected chickens from the less virulent *meq* isoform viruses (vacMeq and vMeq; Fig 4C). On the other hand, the higher virulent *meq* isoform viruses were able to overcome the

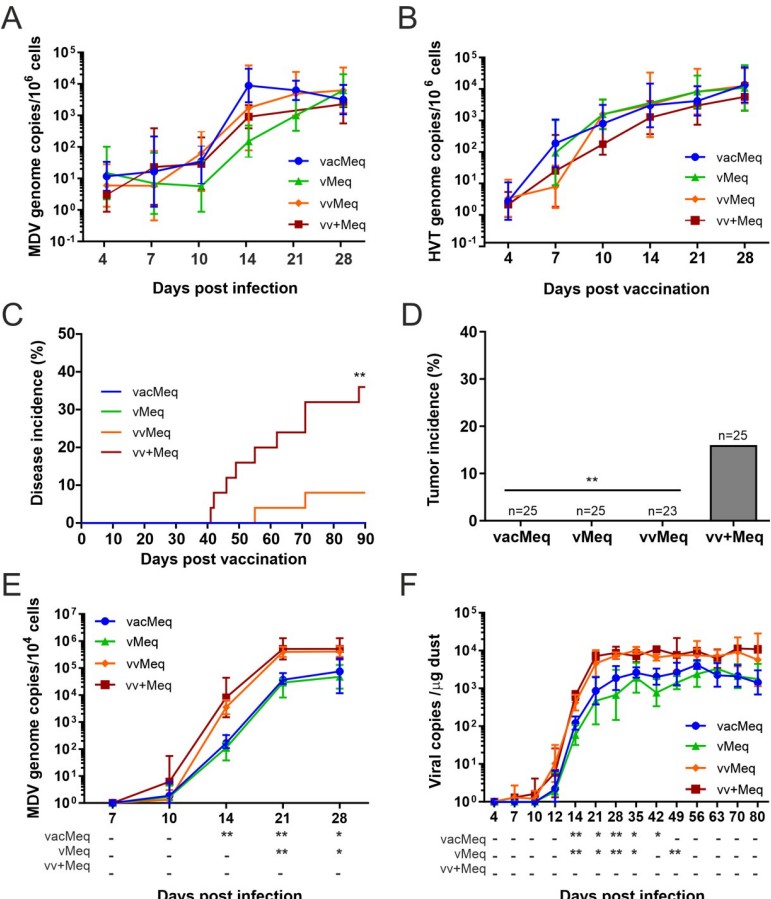

**Fig 4. Pathogenesis and shedding of different *meq* isoform viruses in vaccinated chickens.** Viral genome copy numbers of (A) the *meq* isoform viruses and (B) the HVT vaccine detected in blood of vaccinated chickens infected with the *meq* isoform viruses (p>0.05, Kruskal-Wallis test). (C) Disease incidence and (D) tumor incidence in vaccinated chickens infected with indicated recombinant viruses. Asterisks (** p<0.0125, Fisher's exact test) indicate statistical differences to vv+Meq in (D). (E) Viral copies from feathers of the *meq* recombinant viruses. (A), (B) and (E): mean MDV genome copies per one million cells are shown for the indicated time points. (F) Viral copies per μg of dust are shown for each group as validated previously [32]. Statistical differences in the feathers and dust samples are displayed as a comparison to vvMeq. Asterisks indicate significant differences (* p<0.05 and ** p<0.0125; Tukey's multiple comparisons test).

vaccinal protection and caused disease (vvMeq and vv+Meq; Fig 4C). Strikingly, insertion of the vv+Meq isoform strongly enhanced virulence in vaccinated animals (Fig 4C). Only chickens infected with vv+Meq developed tumors (Fig 4D), indicating that the few point mutations in *meq* allow the virus to overcome the vaccinal protection and cause tumors in vaccinated animals.

## Role of *meq* isoforms in virus shedding from vaccinated animals

Efficient virus shedding plays an essential role in virus evolution. During infection, MDV is transported to the feather follicle epithelia in the skin, where it is shed with the feathers into the environment [32].To assess if the *meq* isoforms also affect virus shedding, we collected feathers and dust during the experiment and measured MDV copy numbers by qPCR (Fig 4E and 4F). Even though all viruses reached the feather follicles at approximately ten days post-infection (dpi), virus load was significantly increased in viruses harboring vvMeq and vv+Meq

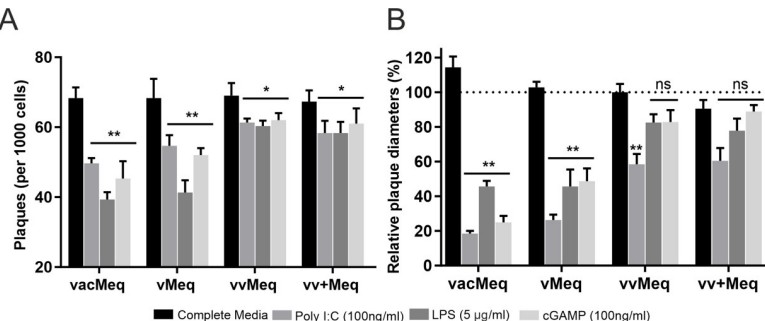

**Fig 5. Efficiency of *meq* isoform viruses to overcome innate immune responses.** Primary T cells were activated by innate immunity agonists (Poly I:C, LPS, or cGAMP). Activated T cells were infected with the different *meq* isoform viruses to determine the effects on virus shedding and replication. (A) Plaque counts were performed on CEC overlaid with 1,000 activated infected primary T cells. (B) Corresponding changes in plaque sizes on infected CEC (normalized to vvMeq). Asterisks indicate significant differences ($^*$ p<0.05 and $^{**}$ p<0.0125; Tukey's multiple comparisons test).

(Fig 4E). In addition, shedding was significantly higher upon infection with the vvMeq and vv+Meq viruses (Fig 4F), indicating that these mutations provide an evolutionary advantage due to the higher virus levels in the environment. Taken together, we could demonstrate that few mutations in *meq* contribute to a higher virulence, allow the virus to overcome vaccinal protection and enhance virus shedding.

## Mutations in *meq* allow the virus to overcome cellular innate responses

To determine if the specific mutations in *meq* affect innate immune responses, we stimulated primary chicken T cells with innate immune agonists (Poly I:C, LPS and cGAMP) and infected these cells with the different recombinant viruses. Upon infection, we measured the effect of these innate immune agonists on virus spread to CEC and subsequent virus replication (Fig 5). Poly I:C, LPS, and cGAMP treatments in general significantly decreased the number of plaques (Fig 5A), and the plaque sizes (Fig 5B) compared to the media control. Strikingly, viruses harboring the higher virulent *meq* isoforms (vv and vv+Meq) formed significantly more plaques than the ones with lower virulent isoforms (Fig 5A). Consistently, CEC infections with higher virulent *meq* isoform viruses led to increased plaque sizes compared to vacMeq and vMeq (Fig 5B). These results indicate that the mutations in the higher virulent *meqs* allow the virus to overcome innate cellular responses induced by these agonists and provide a potential explanation for the vaccine breaks mediated by *meq* [33].

## Discussion

MDV strains have repeatedly increased in virulence and overcame vaccinal protection [34,35]. Virulence is a complex trait and several virulence factors act alone or orchestrated with each other to drive pathogenesis and tumor formation. These factors include the oncoprotein Meq, the viral telomerase RNA (vTR), the virus-encoded chemokine vIL-8/vCXCL13, RLORF4, RLORF5a, pp14, pp38 and telomere arrays present at the ends of the virus genome [6,36]. In this study, we determined the contribution of *meq* isoforms alone in MDV pathogenicity, oncogenicity, and shedding in unvaccinated and vaccinated animals. We provide the first experimental evidence that distinct polymorphisms in the *meq* have a substantial impact on the evolution of MDV towards greater virulence. Our data revealed that only four amino acid changes (AKQV) are involved in an increase in tumor incidence by more than 50% in our experiments.

We first evaluated the growth properties of the *meq* isoforms *in vitro* and *in vivo* to determine if *meq* isoforms from different pathotypes affect virus replication. The *meq* isoforms did not differ in their replication properties in tissue culture and in the host. Even though Meq is expressed during lytic infection, these few mutations in *meq* do not provide an advantage for its replication properties. Consistently, Lupiani and colleagues previously demonstrated that *meq* is dispensable for virus replication [21]. We demonstrate that the minor mutations residing in the *meq* isoforms did not affect *meq* expression in primary CEC (Fig 1E). In addition to the Meq protein, alternative splicing gives rise to a splice form with exon 2 and 3 of vIL-8, designated as *meq*/vIL8 [37]. We assessed the expression of this splice variant by qRT-PCR in both CEC and CD4 T cells, revealing that these minor changes in *meq* do not affect *meq*/vIL8 splicing (Fig 1F). This is consistent with a previous study that showed that splice variants did not differ between different pathotypes in infected primary chicken B cells [38]. The comparable expression of *meq*/vIL8 likely due to the absence of mutation in the splice donor site encoded in the leucine zipper domain in the *meq* isoforms, while the branch point and acceptors sites are outside of *meq* and were not altered in our study.

Deletion of *meq* led to an abrogation of tumor formation, indicating that *meq* has essential transforming properties [39]. The observed increase in virulence of strains over the years has been characterized by the ability to induce lymphoproliferative lesions [13] and an increase in shedding [5], thereby shifting our focus towards these aspects and the contribution of *meq*.

In the first animal experiment, we infected one-day-old chickens with viruses harboring *meq* isoforms from different pathotypes to determine their individual contribution to virus-induced pathogenesis and oncogenesis *in vivo*. In this experiment, we also co-housed the infected with naïve contact chickens to measure the horizontal spread via the natural route of infection. The *meq* gene from the lowest virulence class, vacMeq, completely abrogated MDV pathogenicity and tumor formation. It has been previously shown that the *meq* isoform of the CVI988/Rispens vaccine, is a weaker transactivator, decreasing the expression of cellular and viral genes due to mutations in the DNA binding domain at positions 71 and 77 (Fig 1A) [40]. Meq binds to its own promoter and through its weak transactivation properties on its own promoter it could alter the development of T cell tumors. However, we did not observe a reduction in vacMeq expression on our experiments. The two point mutation differences in vacMeq ultimately rendered the very virulent RB-1B strain apathogenic (Fig 2). Insertion of the vMDV *meq* into RB-1B reduced disease incidence and tumor incidence in infected chickens. The vMDV *meq* (JM/102W) harbors a 177 bp insertion or duplication of a proline-rich (PRR) domain [40] located in the transactivation domain (Fig 1A). This insertion increased the copy number of the PRR, which exerts a transrepression effect [41,42]. The higher virulent forms vvMeq and vv+Meq showed higher disease incidence rates and enhanced oncogenesis compared to the less virulent pathotypes (Fig 2B–2D). An independent animal experiment using a different chicken line confirmed the markedly elevated disease incidence (S1A Fig) and the higher oncogenic potential for the higher virulent *meq* isoform viruses (S1B Fig). The vv+Meq had a slightly lower disease incidence than vvMeq (Figs 2B and S1A). This could be due to epistatic effects, where the fitness of the virus is impacted not by *meq* alone, but by its interaction with the rest of the viral genome. Interestingly, this effect was not detected upon natural infection in contact animals (Fig 3B). Kumar and colleagues previously inserted *meq* of RB-1B into rMd5 (both vv strains), in which the *meq* only differs in three amino acid positions. This exchange altered the phenotype of the resulting virus in the subtle way and allowed the establishment of tumors cell lines (UD36-38) which could not be achieved with the parental rMd5 [41]. Tumors induced by the recombinant virus showed similar cellular expression profiles to rMd5 tumors, suggesting that the context of the strain encoding the Meq protein plays an important role in pathogenesis. Potential epistatic effects are a limitation in our study and it

remains to be addressed whether different backbones expressing the *meq* isoforms might behave differently.

All recombinant viruses were successfully transmitted to contact chickens (Fig 3A), but only contact chickens in the higher virulent *meq* isoform groups showed clinical signs and tumors (Fig 3B and 3C). The tumor dissemination was also altered upon insertion of the different *meq* isoforms. While the vMeq tumors were only localized in one organ (spleen), multiple organs were affected with the higher virulent *meq* viruses (Fig 2D). We found the highest number of tumors in the vv+Meq group (Fig 2D) and observed the same trend of tumor dissemination in the contact chickens (Fig 3D). Importantly, experimentally and contact birds were hatched on the same day and housed together for the duration of the experiment. Therefore, the contact animals were infected much later (~ day 14) when they were already more resistant to MDV. However, our results clearly show that the higher virulent *meq* isoforms allow tumor formation in more organs in unvaccinated hosts.

In the next animal experiment, we aimed to assess the ability of the different recombinant viruses to break the vaccinal protection and promote efficient horizontal spread. We vaccinated chickens with the HVT vaccine that protects chickens from vMDV (Fig 1A). We then challenged the chickens at day seven post-vaccination using the viruses that harbor the different *meq* isoforms. All viruses replicated efficiently in the vaccinated chickens (Fig 4A and 4B). We observed no mortalities in groups infected with the less virulent *meq* viruses, as observed with the parental strains that cannot overcome the HVT protection (Fig 4C).

The only birds that succumbed to disease despite vaccination were the birds challenged with the higher virulent *meq* isoforms (Fig 4C). However, only the virus harboring the vv +Meq was able to induce tumors in the vaccinated animals. It is remarkable that the virus only required five distinct point mutations in the vv+Meq, allowing the vv+Meq to overcome vaccinal protection and cause malignant tumors (Fig 4D). All of these mutations found in vv+Meq reside in the transactivation domain and affect the number of PRRs. Since the PRRs exhibit a transrepression effect, the mutations interrupt the number of PRRs and thereby influence the transactivation activity of Meq [43]. Moreover, Meq functions in target cellular and viral gene transactivation and the higher transactivation properties of vv+Meq could alter and increase proliferation, mobility and apoptosis resistance of cells that develop tumors, perhaps through the upregulation of adhesion molecules via vTR [44,45]. In addition, the chicken CD30, which is discussed to be involved in MDV lymphomagenesis, has 15 potential binding sites for Meq [46]. Thus, the enhanced transactivation of vv+Meq could also lead to CD30 overexpression, favoring neoplastic transformation. The latter hypothesis is consistent with observation on other oncogenic viruses such as Epstein-Barr virus and Kaposi's sarcoma-associated herpesvirus [47]. However, CD30 overexpression in MDV-induced tumors could not be confirmed in follow-up studies [48].

Efficient virus transmission provides strong evolutionary advantages [49]. Here we found that the mutations in *meq* had a strong influence on the amount of virus presence in the feather follicles and on viral shedding into the environment. The higher virulent *meq* isoform viruses were detected at higher levels in feather follicles compared to the less virulent *meq* isoforms (Fig 4E). Consequently, the levels of virus shedding of the higher virulent *meq* isoforms were increased (Fig 4F), likely providing an evolutionary advantage for the virus. There are two potential reasons for increased virus shedding: i) that the viruses harboring the higher virulent *meq* isoforms replicate better in the feather follicles or ii) that the increased number of transformed cells that can travel to the skin facilitate a more efficient delivery to the feather follicles, enhancing virus production and shedding [50]. Read *et al*. recently demonstrated that vaccination with leaky vaccines prolongs viral shedding and onward transmission of vv+MDV strains as the host is kept alive for extended periods [5]. Also, they showed that the cumulative

shedding of less virulent strains is reduced by vaccination, but increased by several orders of magnitude with highly virulent strains [5].

It would be interesting to evaluate virus competition between the *meq* isoforms to determine which virus sheds at higher rates as performed previously by Dunn et al [51]. They show for pathogenically similar (rMd5 and rMd5/pp38CVI) or dissimilar (JM/102W and rMd5/pp38CVI) virus pairs that the higher virulent strains had a competitive advantage over the less virulent strains [51].

The *meq* isoforms we chose are representative of a broad range of viruses and pathotypes [52]. We did test two *meq* isoforms from the vMDV pathotype, JM102 (Figs 1–5) and 617A (S1 Fig) that behaved similar, resulting in lower disease and tumor incidence compared to viruses harboring a vv and vv+ *meq*. However, it would be interesting to test additional *meq* isoforms from the respective pathotypes in future studies.

Nonetheless, our data indicate that the minor mutations in *meq* contribute to this enhanced shedding that increases the level of infectious virus in the environment and provides a selective advantage for more virulent strains.

Next, we turned to the first line of defense against MDV, the innate immunity. It has been previously shown that Meq blocks apoptosis and interferes with antiviral activity [53,54]. As Meq regulates viral and host genes, we evaluated whether the individual *meq* isoforms affect cellular innate immune responses. The lower virulent *meq* isoforms showed a significant reduction in growth and plaque sizes in cells treated with the agonists (Fig 5). In contrast, the higher virulent *meq* isoforms allow the virus to overcome the antiviral response activated in primary T cells stimulated by Poly I:C-, LPS- and cGAMP (Fig 5). It has been previously shown that MDV has the ability to evade the cGAS-STING DNA sensing pathway (stimulated by cGAMP) as Meq delayed the recruitment of TANK-binding kinase one and (interferon) IFN regulatory factor 7 (IRF7) to the STING complex, thereby inhibiting IRF7 activation and IFN-β induction [33]. Especially the vv and vv+*meq* isoforms were able to block the cGAS-STING DNA sensing pathway, as compared to the lower virulent *meq* isoforms (Fig 5). It remains unclear how Meq mechanistically modulates the signaling pathway and should be investigated to understand the role of Meq in the innate immunity in the future. Overall, our findings suggest that the mutations in the higher virulent *meq* isoforms provide an advantage in the vaccinated animals by allowing the virus to overcome these innate responses early upon infection.

In summary, our data demonstrate that minor polymorphisms in *meq* drastically alter disease outcomes in naïve and vaccinated chickens. The *meq* isoforms from highly virulent MDV strains are required for efficient disease and tumor formation, while those from less virulent strains severely impair or abrogate disease and tumor incidence. Also, we show that the mutations that arose in the *meq* from higher virulent strains permitted vaccine resistance and the ability to shed at higher rates in the environment; all factors promote the evolution of this pathogen.

## Materials and methods

### Ethics statement

All animal work was conducted in compliance with relevant national and international guidelines for care and humane use of animals. Animal experimentation was approved by the Landesamt für Gesundheit und Soziales in Berlin, Germany (approval numbers G0294-17 and T0245-14) and the Agricultural Animal Care and Use Committee protocol (64R-2019-0, UBC protocol 16–023).

### Cells and viruses

CEC were prepared from 11-day old specific-pathogen-free (SPF) chicken embryos (VALO BioMedia, Germany) as described previously [55]. CEC were cultured in Eagle's minimal

essential medium (MEM; PAN Biotech, Germany) supplemented with 10% fetal bovine serum and antibiotics (100 U/mL penicillin and 100 μg/mL streptomycin). Reticuloendotheliosis virus-transformed T cells (CU91) were propagated in RPMI 1640 media (PAN Biotech, Germany) supplemented with 1% sodium pyruvate, 1% nonessential amino acids, 10% FBS, and penicillin–streptomycin, and maintained at 41˚C in a 5% $CO_2$ atmosphere. Viruses were reconstituted by transfecting bacterial artificial chromosome (BAC) DNA into CEC as described previously [55]. Viruses were propagated on CEC for four passages thereafter virus stocks were frozen in liquid nitrogen and titrated on CEC as described previously [56,57].

### Generation of recombinant viruses

To generate recombinant viruses that harbor *meq* isoforms from the different pathotypes, we inserted the *meq* isoforms into the very virulent RB-1B strain (GenBank accession no. MT797629) instead of the native *meq* gene as described previously [28]. This resulted in the viruses containing the *meq* isoforms from CVI988/Rispens vaccine (vacMeq), vMDV strain JM/102W (vMeq), vvMDV strain RB-1B (vvMeq) and vv+MDV N-strain (vv+Meq). Primers used for mutagenesis are listed in Table 1. Insertions of the *meq* genes were confirmed by

**Table 1. Primers and probes used for construction of recombinant viruses, DNA sequencing and qPCR.**

| Construct/target | Primer or probe[a] | Sequence (5'– 3')[b] |
|---|---|---|
| *meq* kana_in (transfer construct) | for | AATTCGAGATCTAAGGACTGAGTGCACGTCCCTGTAGGGATAACAGGGTAATCGATTT |
| | rev | GTCCTTAGATCTCGAATTTCCTTACGTAGGGCCAGTGTTACAACCAATTAACC |
| Δ*meq* (deleting RB-1B *meq*) | for | CAGGGTCTCCCGTCACCTGGAAACCACCAGACCGTAGACTGGGGGGACGGATCGTCAGCGGTAGGGATAACAGGGTAATCGATTT |
| | rev | GGGCGCTATGCCCTACAGTCCCGCTGACGATCCGTCCCCCCAGTCTACGGTCTGGTGGGCCAGTGTTACAACCAATTAACC |
| MDV_*meq* (insertion of *meq*s) | for | ATGTCTCAGGAGCCAGAGCC |
| | rev | GGGTCTCCCGTCACCTGG |
| | for | CGTGTTTTCCGGCATGTG |
| meq/vIL8 (RT-PCR) | for | GCAGGGCGCAGACGGACTA |
| | rev | TCAAAGACAGATATGGGAACC |
| | for | CGTGTTTTCCGGCATGTG |
| ICP4 (qPCR) | rev | TCCCATACCAATCCTCATCCA |
| | probe | FAM-CCCCCACCAGGTGCAGGCA-TAM |
| *meq* (qPCR) | for | TTGTCATGAGCCAGTTTGCCCTAT |
| | rev | AGGGAGGTGGAGGAGTGCAAAT |
| | probe | FAM-GGTGACCCTTGGACTGCTTACCATGC-TAM |
| HVT-SORF1 (qPCR) | for | GGCAGACACCGCGTTGTAT |
| | rev | TGTCCACGCTCGAGACTATCC |
| | probe | FAM-AACCCGGGCTTGTGGACGTCTTC-TAM |
| iNOS (qPCR) | for | GAGTGGTTTAAGGAGTTGGATCTGA |
| | rev | TTCCAGACCTCCCACCTCAA |
| | probe | FAM-CTCTGCCTGCTGTTGCCAACATGC-TAM |
| GAPDH (RT-PCR and qPCR) | for | GAAGCTTACTGGAATGGCTTTCC |
| | rev | GGCAGGTCAGGTGAACAACA |
| | probe | FAM-CTCTGCCTGCTGTTGCCAACATGC-TAM |

[a]for, forward primer; rev, reverse primer.

[b]FAM, 6-carboxyfluorescein; TAM, TAMRA.

PCR, restriction fragment length polymorphism (RFLP), Sanger- and Illumina MiSeq sequencing with a ~1000-fold coverage to ensure that the entire virus genome is correct. The GenBank accession numbers for each *meq* isoform and resultant recombinant viruses can be found in S1 Table.

### Plaque size assays

Replication properties of the recombinant viruses were analyzed by plaque size assays as previously described [58]. Briefly, one million CEC were infected with 100 plaque-forming units (pfu) of the recombinant viruses and cells were fixed at five dpi. Images of randomly selected plaques (n = 50) were captured and plaque areas were determined using Image J software (NIH, USA). Plaque diameters were calculated and compared to the respective control.

### *In vitro* replication

*In vitro* replication of recombinant viruses was measured over six days by qPCR as previously described [59,60]. Briefly, primers and probes specific for MDV-infected cell protein 4 (ICP4) and chicken inducible nitric oxide synthase (iNOS) genes were used (Table 1). The qPCR analysis was performed using an ABI Prism 7700 Sequence Detection System (Applied Biosystems Inc., USA) and the results were analyzed using the Sequence Detection System v.1.9.1 software. Virus genome copies were normalized against the chicken iNOS gene as published previously [50].

### Quantitative reverse transcription PCR (RT-qPCR) and RT-PCR

To assess the expression levels of the *meq* isoforms we performed RT-qPCR as previously described [61]. Briefly, total RNA was extracted from virus-infected CEC and CU91 using the RNeasy Plus minikit (Qiagen) according to the manufacturer's instructions. The samples were treated with DNase I (Promega), and cDNA was generated using the High-Capacity cDNA reverse transcription kit (Applied Biosystems).

ICP4 and GAPDH were used to control for the infection levels and the number of cells (S3 Fig). *meq* expression levels were normalized to the expression levels cellular GAPDH (per million GAPH copies). The primers and probes used for RT-qPCR are shown in Table 1. To investigate the expression of the *meq*/vIL8 splice form in cells infected with the recombinant viruses, we performed RT-qPCR using primers specific for the *meq*/vIL8 splice variant as previously described [57].

### *In vivo* characterization of recombinant viruses

**Animal experiment 1 (pathogenesis of recombinant viruses).** One-day old VALO SPF chickens (VALO BioMedia) were randomly distributed into four groups and housed separately. Chickens were infected subcutaneously with 4,000 pfu of vacMeq (n = 25), vMeq (n = 23), vvMeq (n = 24) and vv+Meq (n = 25). With each group, 11 non-infected contact animals, same age, were housed to assess the natural transmission of the respective viruses. The experiment was performed in a blinded manner to avoid bias. Animals were kept under a 12 h light regime in stainless steel cages with wood and straw litter. Enrichment was provided by perches, sand baths and picking stones. Rooms were air-conditioned and temperature was regulated starting from an air temperature of 28˚C on day 1 decreasing to 20˚C on day 21. In the first 10 days, heat lamps were provided. Food and water were provided *ad libitum*. Whole blood samples were collected for infected animals at 4, 7, 10, 14, 21 and 28 dpi and for contact animals at day 21, 28, 35 and 42 to measure virus load in the blood. The chickens were assessed

every day to monitor for MDV-specific clinical symptoms that include severe ataxia, paralysis, torticollis and somnolence. If symptoms appeared, chickens were humanely euthanized and examined for gross tumor lesions. Tumors were also assessed in chickens that did not show Marek's disease signs upon termination of the experiment at 85 dpi. DNA was isolated from spleens and tumors to confirm the sequence of the inserted *meq* gene and integrity of the viral genome. The phenotypes of the *meq* isoforms were confirmed in a second, independent animal experiment. White leghorn chickens (Sunrise Farms, Inc., Catskill, NY) were inoculated with 1,000 PFU of the respective recombinant viruses (n = 18).

**Animal experiment 2 (infection of vaccinated animals).** One-day old VALO SPF chickens were randomly distributed into four groups as described for animal experiment 1. Chickens were subcutaneously vaccinated with 4,000 pfu of the HVT vaccine (strain FC 126; Poulvac; Zoetis Inc., USA) for each group of 25 chickens. At seven days post-vaccination, chickens were challenged with 5,000 pfu of vacMeq (n = 25), vMeq (n = 25), vvMeq (n = 23) and vv+Meq (n = 25) and similar experimental procedures were followed as in animal experiment 1. Whole blood samples were collected to measure virus load in the blood as described above. Feathers were collected at 7, 10, 14, 21 and 28 dpi to monitor the time and the concentration of the viruses that reached the feather follicles to be shed into the environment. Dust shed from the infected chickens was collected from filters of each room once a week to assess the shedding rates until termination of the experiment at 90 dpi. DNA was isolated from spleens and tumors to confirm the sequence of the inserted *meq* genes.

## Extraction of DNA from blood, feathers and dust

DNA was isolated from blood samples of infected and contact chickens using the E-Z96 blood DNA kit (OMEGA Biotek, USA) according to the manufacturer's instructions. Feathers were collected from birds and the proximal ends of each feather containing the feather pulp (referred to as feather tip). In addition, dust samples (three 1-mg aliquots) were collected from the filters in each room at indicated time points. DNA was extracted from feathers and dust samples as previously described [62]. All samples were analyzed by qPCR. The primers and probes (Table 1) for the differential quantification between MDV and HVT were described previously [63,64]. Briefly, the *meq* gene and SORF1 that are exclusively encoded in MDV and HVT respectively were used as targets in the qPCR.

## DNA extraction from organs and tumor tissue

The innuSPEED tissue DNA Kit (Analytik Jena) was used to extract DNA from organs, according to the manufacturer's instructions. Briefly, 50 mg of tissue were homogenized. The homogenate was treated with RNase A and proteinase K digestion, with the exception to the protocol, that proteinase K treatment was extended to 90 min to release viral DNA from the nucleocapsids. The lysate was cleared by addition of a protein-denaturing buffer following high speed centrifugation. The DNA in the supernatant was isolated on DNA binding columns. After subsequent washing steps, the DNA was eluted in 150 µl elution buffer and used for qPCR or next-generation sequencing analyses.

## Next-generation sequencing of recombinant viruses

DNA sequencing of the recovered viruses and DNA from tumors and spleens were performed on an Illumina MiSeq platform as previously described [65]. Briefly, one to five micrograms of total DNA extracted were fragmented to a peak fragment size of 500–700 base pairs (bp). The fragmented DNA (100 ng to 1 µg) was subjected to next-generation sequencing library preparation using the NEBNext Ultra II DNA Library Prep Kit for Illumina platforms (New England

Biolabs). The bead-based size selection step was performed with Agencourt AMPure XP magnetic beads (Beckman Coulter Life Sciences) selecting for inserts of 500–700 bp. To achieve a library yield >500 ng, five PCR cycles were performed.

We used a tiling array method to enrich the viral sequences from the DNA extracts that were harvested from organs or tumors that contained mainly sequences of chicken origin [65]. The array contained 6,597 biotinylated RNA 80-mers that were designed against the sequence of the RB-1B strain (MYcroarray; Arbor Biosciences). The enrichment was performed according to the manufacturer's instructions.

### Next-generation sequence data analysis

All Illumina reads were processed with Trimmomatic v.0.36 [66] and mapped against the RB-1B strain using the Burrows-wheeler aligner v.0.7.12 [67]. The single nucleotide polymorphism (SNPs) were assessed with FreeBayes v.1.1.0–3 [68]. The data were merged by position and mutation using R v.3.2.3. The SNPs were additionally assessed and generated using Geneious R11 software.

### Quantification of virus genome copies

MDV genome copy numbers were determined by quantitative PCR (qPCR) with primers and probes specific for either the HVT vaccine or *meq* isoform recombinant viruses, to distinguish between the viruses from vaccination and infection (Table 1). Virus genome copies were normalized against the chicken iNOS gene as published previously [50]. The qPCR analysis on feathers and dust was performed as described previously [5,69]. Briefly, for the feather tip samples, viral DNA copies were quantified as genomes per $10^4$ feather tips and for dust, genomes per microgram of dust (MDV genomes/mg dust; based on the mass of dust used to prepare DNA and the volume of dust DNA used per reaction).

### Assessment of virus spread and replication upon treatment with innate immune agonists

Next, we determined if *meq* isoforms allow the virus to overcome cellular innate immune responses in primary T cells. Primary T cells were extracted from the thymus of 12-day old chickens as previously described [70]. T cells were stimulated with either LPS (5 µg/ml), Poly I:C (100 ng/ml), and cGAMP (100 ng/ml), and control (medium only) to induce innate immune responses. At six hours (h) post-activation, T cells were infected with the different *meq* isoform viruses harboring a GFP reporter by co-cultivation with infected CEC due to the strict cell-associated nature of MDV. At 24 h post-infection, viable infected GFP-expressing T cells were isolated by FACS, and 1,000 infected cells were seeded on a fresh CEC monolayer. The number of plaques and plaque sizes were determined at five dpi as described above.

### Statistical analyses

Statistical analyses were performed using Graph-Pad Prism v7 (GraphPad Software, Inc., USA) and the SPSS software (SPSS Inc., USA). The multi-step growth kinetics were analyzed with the Kruskal–Wallis test. Analysis for plaque size assays included a one-way analysis of variance (ANOVA). Kaplan-Meier disease incidence curves were analyzed using the log-rank test (Mantel-Cox test), and Fisher's exact test was used for tumor incidences and distribution with Bonferroni corrections on multiple comparisons. Tukey's multiple comparisons test was used for the analysis of feather and dust samples and for the innate immunity experiments. Data were considered significant if p<0.05.

## Supporting information

**S1 Fig. Pathogenesis in animals infected with *meq* recombinant viruses.** (A) Disease incidence of chickens infected with the indicated recombinant viruses and (B) tumor incidence as percentage of animals that developed tumors during the experiment. Asterisks indicate significant differences compared to vvMeq (* p<0.05 and ** p<0.0125; Fisher's exact test).
(TIF)

**S2 Fig. Next-generation sequencing of recombinant viruses.** (A) The recombinant BACs generated only harbored the natural mutations in *meq* of the different *meq* isoforms inserted in the RB-1BΔIR$_L$. (B) The recovered recombinant viruses in cell culture (passage 4) had no secondary mutations in the genome. Both copies of *meq* are present, as the IR$_L$ is restored. (C) Three representative samples from each recombinant virus from organs or tumor samples were extracted and sequenced. The sequences were aligned with the respective recombinant virus from passage 4. No mutations were detected in *meq*, and only minor point mutations in the minority of the viruses as summarized.
(TIFF)

**S3 Fig. RT-qPCR analysis *in vitro*.** The viral ICP4 (A) and cellular GAPDH (B) expression levels were used to control for the infections and the number of cells respectively. Viral ICP4 copies (A) and cellular GAPDH (B) were assessed by RT-qPCR and were not statistically different (p > 0.05, Kruskal-Wallis test).
(TIF)

**S1 Table. *meq* genes from different MDV pathotypes and genomic sequences from all viruses used in this study.**
(DOCX)

**S2 Table. Meq protein sequence alignments from infected animals.**
(DOCX)

## Acknowledgments

We thank Amr Aswad for careful reading of the manuscript, Ann Reum and Yu You for outstanding assistance, and the animal caretakers for excellent support during our animal studies.

## Author Contributions

**Conceptualization:** Andelé M. Conradie, Mark S. Parcells, Benedikt B. Kaufer.

**Formal analysis:** Andelé M. Conradie.

**Funding acquisition:** Benedikt B. Kaufer.

**Investigation:** Andelé M. Conradie, Luca D. Bertzbach, Jakob Trimpert, Joseph N. Patria, Shiro Murata.

**Methodology:** Andelé M. Conradie, Luca D. Bertzbach, Joseph N. Patria.

**Resources:** Benedikt B. Kaufer.

**Software:** Andelé M. Conradie.

**Supervision:** Benedikt B. Kaufer.

**Visualization:** Andelé M. Conradie.

**Writing – original draft:** Andelé M. Conradie, Luca D. Bertzbach.

**Writing – review & editing:** Andelé M. Conradie, Luca D. Bertzbach, Mark S. Parcells, Benedikt B. Kaufer.

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
