## [Decision Letter · Decision Letter 0]

18 Jun 2020

Dear Dr. Kaufer,

Thank you very much for submitting your manuscript "Few polymorphisms in a single herpesvirus gene enhance virulence and mediate vaccine resistance" for consideration at PLOS Pathogens. As with all papers reviewed by the journal, your manuscript was reviewed by members of the editorial board and by several independent reviewers. In light of the reviews (below this email), we would like to invite the resubmission of a significantly-revised version that takes into account the reviewers' comments.

Thank you for submitting your study to PLoS Pathogens. After three careful and detailed independent reviews, there are a number of substantial concerns about the claims in the manuscript as written. The comments have merit and raise appropriate questions that I would recommend you carefully consider and incorporate into future versions of the text. Attention to these points will improve the manuscript.

We cannot make any decision about publication until we have seen the revised manuscript and your response to the reviewers' comments. Your revised manuscript is also likely to be sent to reviewers for further evaluation.

Sincerely,

Moriah L Szpara

Guest Editor

PLOS Pathogens

Erik Flemington

Section Editor

PLOS Pathogens

Kasturi Haldar

Editor-in-Chief

PLOS Pathogens

orcid.org/0000-0001-5065-158X

Michael Malim

Editor-in-Chief

PLOS Pathogens

orcid.org/0000-0002-7699-2064

Thank you for submitting your study to PLoS Pathogens. After three careful and detailed independent reviews, there are a number of substantial concerns about the claims in the manuscript as written. The comments have merit and raise appropriate questions that I would recommend you carefully consider and incorporate into future versions of the text. Attention to these points will improve the manuscript.

Reviewer's Responses to Questions

**Part I - Summary**

Reviewer #1: This is a very straightforward paper where the authors investigated the long-believed association of Meq with MDV virulence. Using Red-mediated recombination and the infectious RB1B BAC clone, they inserted Meq genes from MD vaccine (Rispens) and three different MDV strains with defined virulence (v – JM/102W; vv – RB1B; vv+ - MK). These recombinant viruses were tested for pathogenicity and the ability to overcome HVT vaccinal protection. The results were the most surprisingly clean (especially Fig. 2B) that I’ve ever seen for MDV when birds are involved. And to my knowledge, this is the first MDV paper that associates more than one variant to disease incidence.

The authors took one next logical step, which was to determine what mechanisms might be involved in increased virulence. Using T cells that were stimulated by various agents, they convincing demonstrate that more virulent Meq isoforms can overcome the innate response that induce pathways involving TLRs or STING.

The main limitation is my opinion is that given the status of this journal, I would have liked to seen more experiments. Easy ones that come to mind include:

• Biological replicates. This is traditional for MDV experiments given the variation that is normally observed by everyone else. I didn’t put this down as a Major Revision but…

• Competition assays, i.e., infect birds with two or more rMDVs to prove that the more virulent one does dominate, shed earlier and faster, etc. In my opinion, this is more definitive proof.

• Looking at the relative expression of Meq and especially the Meq/vIL8 splice variant. So do the variants influence how much and what isoforms are expressed?

• Analyzing which T cell population is being infected by each rMDV and addressing whether there are differences.

Finally the Discussion. This section just summarized the Results and not much else. So the lack of depth, speculation on how the Meq isoforms mechanistically affect virulence, etc. was disappointing especially considering the groups involved. Also as I recall seeing, the Parcells group has made a number of rMDVs with other forms of Meq that did not behave as expected. So while we always want to portray a positive picture, biology is never simple, so the authors should give a more balance discussion that includes potentially conflict results, the role of Meq/vIL8, acknowledge that there are probably other genes involved in the complex trait of virulence, etc.

Reviewer #2: The authors swapped out the meq genes from various pathotypes of MDV-1 and placed them in the RB1B (vv) backbone. They tested their recombinants for replication in vitro and in vivo, pathogenicity, tumor dissemination, and shedding. The strength of the manuscript is that the science is sound. The significant finding is that recombinants containing higher virulent meqs overcome innate immunity in vaccinated animals and shed better than those with lower virulent meqs. The research is proper but the manuscript is not worthy of the study and has numerous shortcomings.

Reviewer #3: In the manuscript by Conradie et al., the authors explore the idea that a single gene (the meq gene) may be largely responsible for the evolution of virulence (and the evolution of vaccine escape) that was observed over the past 50 years in a chicken pathogen, Marek’s disease virus (MDV). To explore this question, they make four recombinant viral strains that differ only in their meq gene, with each receiving either a vaccine type “vac”, a low virulence type “v”, a high virulence type “vv”, or a hyper virulence type “vv+”. They then perform in vitro and in vivo studies to show that the strains have different phenotypes, largely corresponding to what would be expected based on their type alone. Given that the meq gene is the only difference between these strains, the authors conclude that the meq gene is largely responsible for virulence and vaccine escape in the MDV system. This manuscript represents a large amount of work. The experiments are quite elegant, and the results are interesting. That said, I have two substantial issues that I think the authors need to directly address, before this would be appropriate for publication. In light of these two issues, I feel that the authors should probably also weaken some of their strongest statements (for example, lines 96-99 and 258-260).

**Part II – Major Issues: Key Experiments Required for Acceptance**

Reviewer #1: As I wrote above, I was a bit disappointed in the Discussion. Not really sure if this qualifies but as a Major Issue but since it does not involve more experimentation, I think this limitation needs to be elevated especially since the status of the journal.

Reviewer #2: The weakness is the discussion or lack thereof. The discussion is mostly a summary of the results. There is no whole-genome sequencing data in the manuscript, nor relevant accession #s. There is no MK (vv+) sequence data available, and frankly, I believe they used the vv+ strain know as TK.

If the authors are not going to examine revertants, then they must provide whole-genome sequencing data. A proper paper would contain whole-genome data for virus stocks before animal studies and whole-genome data of virus from primary infected and contact birds (without or limited propagation in tissue culture).

Here are some issues that will make the manuscript publishable.

Major.

First of all, the title is not very scientific. What are a few? Higher than two but less than a handful? How about 4 (amino acids A, K, Q, and V)?

Lines 61 through 67. This is not informative to non-Marek's disease scientists. Please define what first, second, and third generation vaccines are. Turkey herpesvirus, GaHV-2, CVI988 (Rispens), etc.

Line 83. There is no MDV-1 strain called MK in the cited paper. Do you mean TK? Please give a GenBank accession # for the MK strain.

Lines 109-111. Where are the accession numbers? Where is the sequencing data? Did all the recombinants have the same sequences outside the Meq loci?

Lines 143-145. vv meq is more virulent than vv+ meq. Why? This was not mentioned in the discussion. In essence, RB1B containing its ‘normal’ meq is more virulent than RB1B containing a vv+ meq. Isn’t that worth mentioning in the discussion?

Lines 162-164. Four amino acids AKQV involved in a >50% increase in tumors. Shouldn’t this be mentioned?

Lines 172-174. The vmeq RB1B recombinant in primary animals had ~ 10% tumors, but no tumors in vmeq RB1B contact birds? Did, or can the author sequence the MDV genome from the vmeq RB1B contact birds to determine what happened, especially to the meq gene? This was also not mentioned in the discussion.

Line 248-249. Why does LPS affect vac/v differently than vv/vv+ relative to the induction by poly IC and cGAMP? Isn’t that interesting and worthy of mentioning in the discussion.

The discussion is basically a reiteration of the results, only lines 265-272 mention outside information.

Line 268. Lupiani used md5, which is a vv. How does deleting a vv meq not affect its replication in vivo? Isn’t that surprising?

Line 288-290. This is complicated. Is this expansion region found in other meqs of lower and higher virulence? Jm102 has a 59 amino acid expansion, and CVI988 has a 60 aa expansion. Need accession numbers for these sequences you used.

Line 313 -318. This is just as a reiteration of results.

Line 321- 326. How does oncogenic potential affect shedding??? Please come up with a hypothesis or something in the discussion.

The Material and methods lack the passage history to generate the MD stocks. How many times were the reconstituted viruses passed in cell culture before use in animal experiments?

Line 360. Accession number is EF523390.1 is for an RB1B that doesn’t transmit horizontally. Did you use the repaired RB1B BAC? Where is the reference for this?

Where are the results of full genome sequencing? You need this when you don't examine revertants. What about secondary mutations.

Reviewer #3: The first issue is that the authors use a single meq genotype taken from a single MDV isolate to represent the entirety of a pathotype (i.e. vac, v, vv, vv+). While this may be a reasonable assumption for the vaccine type, the other three pathotypes have population-level genetic variation within a single pathotype. Perhaps the authors have very good justification for selecting the strains that they have, but regardless, I am left wondering whether the same pattern would have been observed had they chosen, for example, a different representative “v” isolate. Obviously, it is unreasonable to expect the authors to repeat this analysis with a panel of genotypes, but I do think that this limitation of their study should be discussed.

The second issue is that all of these experiments were conducted in a single genetic background, the RB1B background, and so epistatic effects between meq and the rest of the genome were not controlled for. RB1B is a “vv” virus, and it also happens to be the same virus from which their “vv” meq was chosen. I am thus left wondering how much of their results are due to epistatic effects, where the fitness of the virus is impacted not by meq alone, but by its interaction with the rest of the viral genome. Sequence data show that vaccine isolates and low virulence v isolates tend to cluster together on one branch of a phylogenetic tree while higher virulence vv and vv+ isolates tend to cluster together on another branch. I therefore worry that some of their main conclusions may be the result of a “v” or “vac” meq simply not performing all necessary functions to maintain viral fitness in a “vv” background. Evidence to suggest epistatic effects can be seen in Figure 2A, 2B, and 2C, where the “vv” strain appears to be more virulent than the “vv+” strain, which could be explained by epistatic effects but not by the hypothesis posed by the author that the meq gene controls virulence. The observed low virulence of “v” and “vac” meq strains would be predicted by both the epistatic effect hypothesis and the hypothesis posed by the authors have the same prediction, and so it is difficult to conclusively determine which is responsible for their results. Again, I think it is unreasonable to expect the authors to repeat their set of experiments using a “v” or “vac” background, but I again think that it is necessary to discuss the limitations that result from experimental constraints.

**Part III – Minor Issues: Editorial and Data Presentation Modifications**

Reviewer #1: In no special order, these are the items that I believe would make this a better paper

• Biological replicates. I waffled as to whether this should be a major issue especially since I’ve never seen such clean results with MDV in my life, especially those involving birds.

• Title: Might qualify this a bit something like “Few polymorphisms in a single herpesvirus gene are capable of enhancing virulence and mediate vaccinal resistance.” I say this because I believe that virulence is a complex trait and there are likely many other existing variants beside those in Meq that contribute.

• Multiple places you say “vaccine protection.” I think “vaccinal protection” is more appropriate.

• Lines 36-37. Possibly rephrase as “One virus that should shown repeated shifts to higher virulence in response to more efficacious vaccines has been the ...”

• Line 40. Replace “dictate” with “can significantly alter”

• Line 43. Delete “a” at the end of the line.

• Line 48. You say some viruses but isn’t smallpox the only example?

• Lines 52-29. This doesn't flow well. You just introduced that MDV is a well-known example of viral evolution to greater virulence. But then you give rambling facts. Why don't you first define what is MDV, what is the pathology, and how it has increased in virulence over time?

• Line 63. While this may be true, it has never been really proven. What is true is that vvMDVs appeared ~10 years after the introduction of bivalent MD vaccine. I know that this is picky but "led" implies causative to me. There may have been other factors for increased virulence.

• Line 73, Figure 1A. You give vaccine names but earlier you call them first, second, and third generation vaccines. Try to be consistent.

• Line 74. Replace “remains elusive” with “has never been proven”

• Line 76. Add a comma after “cells” to help separate the phrases

• Line 88. Replace “instead of” with “to replace”

• Entire Results section. Strictly speaking, no introductory material, citations, interpretation, etc should be given in this section.

• Lines 102-111. This should be moved to the Methods section. Also it would be nice to separate the reads, coverage, etc. by Sanger from Illumina.

• Line 117. I think you mean Fig. 1C

• Line 119. Similarly, I think you mean Fig. 1D

• Line 135 and elsewhere. You use what I considered fairly high amounts of viral and vaccine pfu. Why? Is this because the chicken lines you use are genetically resistant? Remember that in pathotyping of MDV strains, ADOL and the vaccine industry use highly susceptible birds.

• Line 137. It might have been good to demonstrate that your qPCR assay worked equally well for all recombinant viruses since it appears you're using the same Meq probe.

• Line 138. Bursal and thymic atrophy are more observed in more virulent MDV strains. Did you see this and if yes, then you should add this comment.

• 153. Are these tumors clonal? Not sure if this matters much but I would think higher virulent strains would lead to more non-clonal tumors.

• Line 172 and Fig. 3B. This is a bit surprising as the vMeq should have caused disease. Also both the vvMeq and vv+Meq gave low disease incidence. Again, could this be due to using MD resistant birds?

• Line 199 and Fig. 4D. A bit odd as HVT should protect only against v MDV strains; remember vv MDV strains are defined as the ability to evade HVT protection.

• Line 224. You could made a stronger and more definitive statement about whether more virulent strains do transmit better is if you had performed mix infections.

• Lines 253. Consider changing “losses” to “costs” as the condemnation rates are very low these days.

• Line 255. Consider changing “Nevertheless, MDV strains increased in virulence and overcame vaccine…” to “MDV strains have repeatedly increased in virulence and overcome vaccinal…

• Line 258. Delete “could”

• Line 268. In fact, I would say deletion of Meq enhanced lytic replication, likely due to the inability of this rMDV to undergo latency.

• Line 280. Change to “positions” (plural)

• Line 288. The correct name is JM/102W

• Line 300. Suggest replacing “dissemination” with “tumor formation”

• Line 318-321. The Read et al. paper is important but one has to realize that the main reason more virulent MDVs shed more in vaccinated birds was that the unvaccinated ones died quickly. So this is what normally happens in the real world. Plus MD vaccines definitely reduce shedding in living birds. So to say that MD vaccines enhance viral shedding is incorrect.

• Line 354. It is traditional to give passage levels for all the viruses.

• Line 392. For the contact birds, were these the same age and introduced into the cages as the original challenge birds. Not clear to me.

Reviewer #2: Minor issues.

Line 68. Use ‘leaky’ This sounds like slang.

Lines 70-71. There should be references for this.

Lines 106-108. This is confusing. You replaced the meq of vv RB1B with a different vv meq. It sounds like you are replacing RB1B meq with RB1B meq.

Lines 116-129. This is not labeled correctly. 1D is plaque size.

Line 181. There is no key for 3a. The x-axis is not continuous time. Some indication of the two groups is needed. Or make two figures.

Line 183. Red and orange problem. Please use more contrasting colors like red and blue.

Line 191. Mention how the chickens were challenged. SQ?

Lines 203-206. It is hard to read these graphs esp 4C. Difference between red and orange. Use purple or light blue?

Line 354. This is not the correct reference on how to rescue an MDV BAC.

Reviewer #3: Smaller issues:

Introduction: I think the word “chickens” doesn’t appear until line 91, which is way after the system is introduced. I have no problem with trying to keep things general, but I do think it is necessary to say that MDV is a pathogen of chickens. For example, it seems like it would fit really well in line 52. “…Marek’s disease virus (MDV), a pathogen of chickens.”

Lines 109-111: Does this analysis ensure that the meq gene was inserted in the correct spot (two spots) and nowhere else? I am not familiar enough with these methods to know, but if so, the authors should say so.

Figure 1A (left side): My brain had trouble reading “low virulence pathotypes” at the top of the figure and “high virulence pathotypes” at the bottom of the figure, especially because that flips on the right side of this figure. Might be worth flipping.

Figure 1A (right side): To my knowledge, vv++ strains have not yet been described, and so the arrow that extends beyond the CVI line is misleading. Perhaps a question mark could be added.

In many places throughout the text (i.e. 140-141, 150-151, 172-174, 277-278, there may be others that I missed), the authors state that the v meq or the vac meq “abrogated” or removed virulence or reduced oncogenicity. However, this wording is misleading, because although RB1B has high virulence, the Delta meq RB1B isolate has low virulence and oncogenicity. Adding the v meq therefore increased virulence and oncogenicity, just not by as much as adding a vv meq.

On lines 145-148, the authors state that the vv+ meq causes neurovirulence. This would be interesting given that vv+ isolates of MDV are typically associated with neurovirulence (suggesting that meq plays a role in neurovirulence) but the data are not presented. The data should either be presented, or the statement should be removed.

Line 176: It should be stated that the “slightly enhanced” effect is not significant.

Figure 3 legend (lines 183-184): I found the wording here to be confusing. Are panels C and D showing data for experimentally-infected birds or for naturally-infected birds. If the latter, why is tumor incidence higher than disease incidence? That brings me to ask how you are defining disease incidence in general (is it mortality and/or reaching clinical endpoints?)? If so, that should be stated.

Figure 3: I was also wondering whether disease incidence is significantly different between pathotypes. I can see that the pattern is in the direction that would be expected based on pathotype, but I cannot determine from the figure whether the vv and vv+ isolates are significantly more likely to cause disease than v or vac isolates.

Line 189 (or possibly in methods): Are these chickens maternal antibody positive or maternal antibody negative.

Figure 4 legend (line 211): Should “vacMeq” be “vvMeq”? Based on the figure it appears that it should. If not, the tables below Fig 4E and 4F are either wrong or I am misinterpreting how to read them.

Figure 5B: Relative plaque sizes differ between the different virus strains, and so correcting for plaque size is a little misleading. The plaques may actually be very similar sizes between vacMeq and vv+Meq in the immune activated treatments, but because of differences in the reference (non stimulated) treatment, it would look like the different meq variants are altering ability to escape innate immunity. I would want to see the raw data based on actual plaque size rather than corrected plaque size.

Line 258: Delete “could”.

Lines 262-264: This sentence needs to be reworded to be grammatically correct.

Lines 281-282: The Rispens vaccine has a large insert. Was this insert included in the “vacMeq”? If so, the sentence on lines 281-282 is misleading. If not, why was the full Rispens meq gene not used? Regardless, you should be clear about the sequence that was inserted. Perhaps full meq genes could be included as supplemental info.

Lines 288-289: Again I feel that this sentence may be misleading unless I am confused. My understanding is that JM102 does not have a 21 amino acid insertion, but rather a series of point mutations that make it differ from typical vv and vv+ isolates. Providing the full sequences for the meq regions inserted is needed.

Lines 294-295: Was this difference statistically significant?

Lines 389-391: There are missing details here. How were animals housed? As a reader, I am curious about their space, airflow, disinfection between experiments, etc. Also, did any animals need to be culled for space reasons, or did any animals die from anything other than MD? If so, how did you deal with this in your statistical analyses?

PLOS authors have the option to publish the peer review history of their article (what does this mean?). If published, this will include your full peer review and any attached files.

Reviewer #1: No

Reviewer #2: No

Reviewer #3: No
---

## [Decision Letter · Decision Letter 1]

15 Sep 2020

Dear Dr. Kaufer,

Thank you very much for submitting your manuscript "Distinct polymorphisms in a single herpesvirus gene are capable of enhancing virulence and mediate vaccinal resistance." for consideration at PLOS Pathogens. As with all papers reviewed by the journal, your manuscript was reviewed by members of the editorial board and by several independent reviewers. In light of the reviews (below this email), we would like to invite the resubmission of a significantly-revised version that takes into account the reviewers' comments.

We appreciate the resubmission of this revised manuscript. However the scope of revisions to the main manuscript did not match the level of detail in the rebuttal. This assessment was shared by 2 of the 3 reviewers, and by the editors. It is indeed necessary to address all of the reviewers points in the manuscript itself, and not simply in the rebuttal. PLoS Pathogens does not have a word limit, and thus there is no editorial or word-limit on the requested clarifications to the text. As editors, we agree with the reviewers' concerns that the manuscript overstates the impact of the findings. In addition, by omitting discussion of limitations that the reviewers requested be included, the manuscript leaves the reader with a grander impression of what has been accomplished. This is a valuable piece of work and a contribution to the field. Please make a concerted effort to incorporate text and explanations from the original rebuttal into the main text. In addition, please attend to the additional points raised by two reviewers.

We cannot make any decision about publication until we have seen the revised manuscript and your response to the reviewers' comments. Your revised manuscript is also likely to be sent to reviewers for further evaluation.

Sincerely,

Moriah L Szpara

Guest Editor

PLOS Pathogens

Erik Flemington

Section Editor

PLOS Pathogens

Kasturi Haldar

Editor-in-Chief

PLOS Pathogens

orcid.org/0000-0001-5065-158X

Michael Malim

Editor-in-Chief

PLOS Pathogens

orcid.org/0000-0002-7699-2064

We appreciate the resubmission of this revised manuscript. However the scope of revisions to the main manuscript did not match the level of detail in the rebuttal. This assessment was shared by 2 of the 3 reviewers, and by the editors. It is indeed necessary to address all of the reviewers points in the manuscript itself, and not simply in the rebuttal. PLoS Pathogens does not have a word limit, and thus there is no editorial or word-limit on the requested clarifications to the text. As editors, we agree with the reviewers' concerns that the manuscript overstates the impact of the findings. In addition, by omitting discussion of limitations that the reviewers requested be included, the manuscript leaves the reader with a grander impression of what has been accomplished. This is a valuable piece of work and a contribution to the field. Please make a concerted effort to incorporate text and explanations from the original rebuttal into the main text. In addition, please attend to the additional points raised by two reviewers.

Reviewer's Responses to Questions

**Part I - Summary**

Reviewer #1: I like the revisions that the authors provided especially the inclusion of NGS to verify that there were no unexpected changes in the viral genome. I also agree with the major conclusions, which are (1) small amino acid changes can have large effects on virulence. (2) Meq isoforms are associated with evading the innate immune response.

Reviewer #2: The is the second review of the manuscript and it is much imptoved especially the discussion section.

Reviewer #3: This manuscript is improved from the previous version I saw, but the two major criticisms that I brought up previously were not adequately address in the text. I should say that I am somewhat annoyed that it feels as though my comments were viewed as obstacles to publication rather than opportunities to improve the text, but spite is not a reason to reject a manuscript and so I hope that the authors will make the suggested changes listed below.

The authors addressed my criticisms in the text of their response letter, but I do not see adequate changes to the text of the manuscript to address these concerns for future readers. In particular, my two major criticisms were that 1) they used only a single version of meq to represent each pathotype, and 2) they used only a single pathogen background and so cannot definitely rule out epistatic effects. With regard to 1, they wrote the following reasonable response but did not include it in the manuscript text. A version of this text needs to be included in the discussion:

"Thanks for giving us the opportunity to discuss this aspect. We selected these v, vv and vv+ genes as they are found in several strains of the respective pathotype. We therefore believe that our data is representative for a broach range of viruses and pathotypes, even though we cannot confirm this for every virus strain and meq sequence. In addition, we recently tested another mutant that harbors an alternative v meq isolate (617A) in collaboration with Dr. Parcells and obtained comparable results (Fig. S1)."

With regard to point 2, the authors wrote that they responded, but I do not see a response. They wrote in the letter: "As suggested, we discussed the potential epistatic effects and the limitations of using one genetic background. These changes improved our discussion section and made our manuscript stronger." However, unless I am missing where they responsed (please include line numbers in response letters in the future), in the text, they do not adequately address this issue. In fact, the only time they use the term "epistatic" is in justifying why the vv+Meq strain does not behave as expected in one case. However, it is critical that they also explain that the possible existence of epistatic effects limits their conclusions.

Also, I asked the reviewers to qualify that one of their observations was nonsignificant. In the previous version that was line 176, and they say in their response letter "Response 10: We now stated this.". In the updated manuscript, this text now appears on lines 164-165. It reads, "In contrast, vv+Meq even showed an enhanced tumor dissemination pattern compared to the very efficient vvMeq (Fig. 2D)." This effect is non-significant. The authors cannot say this without stating that it is not a significant difference. It is very frustrating as a reviewer to have a comment ignored that is this easy to address.

Other comments:

Line 29: "increase" needs to change to "modify" or "decrease" since the authors have only shown that moving from vv to v or vac meq decreases virulence, and not the other way around.

Line 30: Delete "field" since this study wasn't done in the field or done using "field strains".

Line 101-102: This result should be discussed given that "vv" isolates are expected to be able to escape vaccine protection but you did not see that.

Line 154 and 162: "Insertion" is misleading and should be changed. "Replacing the vv meq with v meq" may work better because meq isn't inserted, but rather replaced.

Fig2D: Are you plotting the mean number of tumors or the mean number of tumors given the presence of at least one tumor? You say the former, but I suspect it is the latter and should be fixed.

Line 167: Add "breed" after "chicken".

Line 279: Delete "minor", since "minor polymorphism" has a very specific meaning and it is not the meaning meant here.

Line 307: I do not understand what "and by with weak transactivating properties" means. Please rewrite.

Line 309: "mutations" should be "mutation".

**Part II – Major Issues: Key Experiments Required for Acceptance**

Reviewer #1: Having said that, I would still like to see the following added to the manuscript:

• A more balanced discussion that indicates among others (1) virulence is a complex trait, (2) Meq isoforms do not correlate with virulence of the origin strain, and (3) a comparison of Meq splice variants, especially with vIL8), among the isoforms.

Reviewer #2: The experiments are acceptable.

Reviewer #3: (No Response)

**Part III – Minor Issues: Editorial and Data Presentation Modifications**

Reviewer #1: Title. Should it be “mediating” and not “mediate” as this follows after “capable of”?

Lines 74 and 75. It is “MD,” not “MDV” vaccine. Vaccine is to the disease, not the virus.

Lines 83 and 86. Why do you write “b-ZIP” vs. “bZIP” (no hyphen), which is what I think is more traditional?

Lines 129-131 and Figure 1 (E-G). I’m trying to understand this. Is Figure 1E and !F, the amount of ICP4 and GADPH copies? If so, then how did you determine this? And Figure 1G is relative Meq levels but how did you normalize to both ICP4 and GAPDH (lines 461-463)? I’m confused as when I do relative expression, I do delta delta delta Ct analyses using one normalizing gene. Also to determine the relative expression of a gene, the normalizing gene expression should be relatively static AND close to the level of expression of the test gene to provide sufficient accuracy. I know that this is hard, if not impossible, with an infected cell but if the relative levels are 10e5 or above, then I doubt you have much power in comparing between samples.

Also, given that you performed resequencing of the viral genomes, I would think looking at differences in Meq splice variants, especially the one with vIL8, would not be that difficult.

Lines 164-165. Figure 2D shows that vvMeq has a significantly higher tumor incidence compared to vv+Meq, and no difference in tumor number. Please revise.

Line 167. I think you mean different chicken “line” or “genetics.”

Figures 3 and S1. How can the tumor incidence be higher than the disease incidence for vvMeq and vv+Meq as tumors are one of the main criteria for disease?

Line 216, Figure 4B. It has been repeatedly reported that HVT does not replicate well in chickens, which is contrary to your results. So what controls did you perform to show that your HVT primers to SORF1 do not amplify MDV sequences? One simple test might be to screen contact birds that were co-housed with HVT-vaccinated birds. The thought is that since HVT replicates poorly, it is also known that it rarely transmits horizontally.

Line 282. While the 50+% increase was observed, I think you should provide some moderation by saying something like “in our experiments” as you only tested one background virus.

Lines 292-294. I agree that Meq variants did not affect splicing in B cells but what about CD4 T cells, which are the primary transformation target cell type? You’re pretty much discounting that Meq variants have no influence on splicing, which may out to be true, but not proven. Would like some wording to softening your dogmatic-like claim.

Line 319. You say that the disease incidence for vv+Meq was not significantly lower compared to vvMeq in Figure 2B. However, Figure S1A shows a highly significant difference.

Lines 365-366. Differences in feather load among various recombinant MDV strains appears as early as 10 days and certainly by 14 days. While transformation this early is possible, do you really think that this is the major driving force for the increased viral copies? This is highly speculative. And alternative theory is that different Meq isoforms replicate better in the feather follicles?

Discussion. Still no commentary about how different Meq isoforms lead to varying results. For sure, the authors have demonstrated that their Meq isoforms can lead to higher virulence though not exactly as predicted. This limitation as well as other work from the Parcells group needs to be incorporated. This is not intended to negate the work but rather that it is still difficult to predict the phenotypic outcome of recombinant MDVs even when incorporating variants of a major gene like Meq.

Side comments:

Competition assays do not rely on aa changes and/or antibodies only. One can easily compare the relative amount of viral genomes given known DNA polymorphisms that discriminate between strains.

My question on clonality cannot be addressed by looking at viral genomes. The best method, to my knowledge, is T cell receptor (TCR) spectratyping. Using this assay, you can determine if the tumor was VB1 or VB2, and if the dominant peak are different, then assume that the tumors are not clonal. The point is that more virulent strains might be more equipped to overcome the immune response, which would lead to more viral transformations early on.

If Meq isoforms do not significantly alter viral replication both in vitro and in vitro, yet alter disease and tumor incidence, what does this say about the major mode of activity? In my opinion, the observable effects are on the host genome.

Reviewer #2: Some grammatical errors.

Line 166. The data of this in vivo experiment was validated in an independent animal experiment using a different chicken

Add the word “line” after chicken.

Line 307 Meq binds to its own promoter, and by with weak transactivating properties, the vacMeq could in turn cause low levels of 308 Meq expression in vivo and the development of T cell tumors might fail to occur.

Replace “by with” with the word “through”

Reviewer #3: (No Response)

PLOS authors have the option to publish the peer review history of their article (what does this mean?). If published, this will include your full peer review and any attached files.

Reviewer #1: No

Reviewer #2: No

Reviewer #3: No
---

## [Editor Report · Decision Letter 2]

27 Oct 2020

Dear Dr. Kaufer,

We are pleased to inform you that your manuscript 'Distinct polymorphisms in a single herpesvirus gene are capable of enhancing virulence and mediating vaccinal resistance.' has been provisionally accepted for publication in PLOS Pathogens.

Best regards,

Moriah L Szpara

Guest Editor

PLOS Pathogens

Erik Flemington

Section Editor

PLOS Pathogens

Kasturi Haldar

Editor-in-Chief

PLOS Pathogens

orcid.org/0000-0001-5065-158X

Michael Malim

Editor-in-Chief

PLOS Pathogens

orcid.org/0000-0002-7699-2064
---

## [Editor Report · Acceptance letter]

24 Nov 2020

Dear Dr. Kaufer,

We are delighted to inform you that your manuscript, "Distinct polymorphisms in a single herpesvirus gene are capable of enhancing virulence and mediating vaccinal resistance.," has been formally accepted for publication in PLOS Pathogens.

Best regards,

Kasturi Haldar

Editor-in-Chief

PLOS Pathogens

orcid.org/0000-0001-5065-158X

Michael Malim

Editor-in-Chief

PLOS Pathogens

orcid.org/0000-0002-7699-2064